

# NMR side-chain assignments of the Crimean-Congo haemorrhagic fever virus glycoprotein n cytosolic domain

Louis Brigandat[#1], Maëlys Laux[#1], Caroline Marteau[1], Laura Cole[1], Anja Böckmann[1], Lauriane Lecoq[1], Marie-Laure Fogeron[1], Morgane Callon[1]

[1]MMSB Lyon, UMR5086 CNRS /Lyon University, 7, passage du Vercors, 69367 Lyon Cedex 07, France

[#]These authors contributed equally to this work

*Correspondence to*: Morgane Callon (morgane.callon@ibcp.fr)

## Abstract

We assigned the side-chain resonances of the Crimean-Congo haemorrhagic fever virus (CCHFV) glycoprotein n (Gn) cytosolic (cyto) domain to complete the backbone resonances previously published by Estrada et al. (2011). The process was accelerated by three major factors: first, sample preparation using cell-free protein synthesis (CFPS) was completed in less than 2 days and allowed correct zinc finger formation by adding zinc ions to the reaction. Second, remote access to NMR platforms with standardized pulse sequences allowed data acquisition in 18 days. Third, data analysis using the online platform NMRtist allowed sequential resonance assignments to be made in a day, verified and finalized in less than a week. Our work thus provides an example of how NMR assignments, including side chains, of small and well-behaved proteins can be approached in a rapid routine, at protein concentrations of 150 μM.

## 1. Introduction

The Bunyavirales order is a large group of serologically related, often tripartite, negative-sense RNA viruses. Crimean-Congo haemorrhagic fever virus (CCHFV), a member of the order, is considered one of the most deadly viruses, with a mortality rate of up to 40 % (Ye et al., 2022; Hawman and Feldmann, 2023). CCHFV poses a serious threat to both human and animal health, and is currently spreading to non-endemic areas as their vectors expand their range with global warming. Indeed, the Hyalomma ticks that transmit CCHFV are already present in Europe (Bonnet et al., 2022), and the virus has caused human fatalities in Spain (Lorenzo Juanes et al., 2023); it also has been recently (2023) detected for the first time in southern France.

The three genomic segments of most Bunyavirales can be divided into the small (S), the medium (M), and the large (L) segments. Bunyavirales often share a similar protein composition, consisting of the nucleocapsid (NP) protein, which forms the ribonucleoprotein (RNP) complex with the viral genomes, the two envelope proteins Gn and Gc, and the RNA-dependent RNA polymerase (RdRp). Structural information is abundant for NP (*e.g.* (Sun et al., 2018; Arragain et al., 2019)) and also for RdRp (*e.g.* (Arragain et al., 2019, 2020, 2022)). The Gn and Gc envelope glycoproteins are inserted into the endoplasmic



reticulum (ER) during translation and eventually localize to the surface of the viral particle, forming the trimeric spikes that characterize mature particles. Gn and Gc have a large ectodomain, a transmembrane anchor and a small cytosolic domain (Hulswit et al., 2021). In particular, the Gc ectodomain has been extensively studied (Zhu et al., 2017; Guardado-Calvo and Rey, 2017; Bignon et al., 2019; Mishra et al., 2022). In contrast, its transmembrane and cytosolic domains (Gc$^{TMcyto}$) have not yet been structurally characterized. For some viruses, information is available for the Gn ectodomain (Hastie et al., 2017;

Hulswit et al., 2021). The Gn ectodomain interacts with the one of Gc and undergoes extensive structural changes during maturation (Halldorsson et al., 2018). Several structures exist for the isolated cytosolic domain of Gn (Gn$^{cyto}$), typically from Hanta, CCHF and Junin viruses (Estrada et al., 2009; Estrada and De Guzman, 2011). This domain contains a double ββα zinc finger (ZF) motif that could play a matrix protein role in viral particle assembly (Strandin et al., 2013), as does the Lassa virus Z protein, which alone is sufficient for the release of virus-like particles (Strecker et al., 2003).

The sequential NMR resonance assignments of CCHFV Gn$^{cyto}$ have been reported in the literature, and were deposited in the BMRB under the accession number 17383 (Estrada and De Guzman, 2011). Amide proton and $^{13}$C/$^{15}$N backbone resonances are presented there, alongside with Cβ assignments. However, other $^{13}$C side chain resonances as well as protons remained unassigned. Because of our interest in eventually comparing the isolated domain with the membrane-bound form and its likely multimeric states, which we will address using proton-detected magic-angle-spinning NMR, we now report here

the $^{1}$H, $^{13}$C side-chain resonance assignments, as well as the cell-free sample preparation procedures we used to efficiently prepare the protein domain.

## 2. Materials and Methods

### 2.1 Wheat-germ cell-free protein synthesis (WG-CFPS)

For WG-CFPS, the sequence corresponding to the cytosolic domain of Gn (residues 733-799) was cloned into the pEU-E01-

MCS vector (CellFree Sciences, Japan) with a Strep-tag fused at its C-terminus. The plasmid was then amplified in DH5α cells and purified using a NucleoBond Xtra Maxi kit (Macherey-Nagel, France). Further purification was performed by phenol-chloroform extraction to achieve the required level of plasmid purity. mRNA transcription was performed using 100 ng/μl plasmid, 2.5 mM NTP mix (Promega, France), 1U/μl RNase inhibitor (CellFree Sciences, Japan) and 1U/μl SP6 RNA polymerase (CellFree Sciences, Japan) in transcription buffer containing 80 mM Hepes-KOH pH 7.6, 16 mM magnesium

acetate, 10 mM DTT and 2 mM spermidine (CellFree Sciences, Japan). The solution was incubated at 37 °C for 6 h, and the mRNA produced was used directly for translation. Cell-free synthesis was performed with uncoupled transcription and translation. All steps followed the protocol of Takai and colleagues (Fogeron et al., 2017; Takai et al., 2010). Translation reactions were performed using the bilayer method for 16 h at 22°C. Uniform $^{2}$H-$^{13}$C-$^{15}$N or $^{13}$C-$^{15}$N labeled amino acids were used for sample preparation. The sample was purified by affinity chromatography using a 1-ml *Strep*-Tactin© Superflow©

gravity flow column (IBA Lifesciences). The protein was eluted in 50 mM phosphate buffer pH 6.5, 50 mM NaCl and 2.5 mM desthiobiotin, and concentrated using a 3 kDa Amicon Ultra filter to 109 μM ($^{2}$H-$^{13}$C-$^{15}$N sample) and 150 μM ($^{13}$C-$^{15}$N



sample) as measured using a NanoDrop, and 1x EDTA-free protease inhibitor was added to the final sample to avoid protein degradation. Zinc sulfate was added to each buffer in the manufacturing process at a concentration of 100 µM.

## 2.2 NMR spectroscopy

NMR spectra were acquired at 298 K on Bruker Avance III 600 MHz (3D spectra for backbone assignments) and 950 MHz (3D spectra for side-chain assignments) spectrometers equipped with a cryoprobe. Backbone $^1$H, $^{15}$N and $^{13}$C resonance assignments were performed using standard NMR experiments set up via the NMRlib tool (Favier and Brutscher 2019) in TopSpin 4.0.8 (Bruker Topspin). Specifically, $^{15}$N-HSQC, HNCA, HNCO, HNcaCO, and HNCACB were recorded for backbone assignment, and $^{13}$C-HSQC, HccoNH-TOCSY, hCCH-TOCSY and HCcH-TOCSY were recorded for sidechain

assignment (an $^{15}$N-HSQC was recorded equally on the protonated sample to transfer $^1$H-$^{15}$N chemical shifts to the protonated protein). DSS was used for $^1$H chemical shift referencing. All spectra were processed using TopSpin 4.0.8 (Bruker Biospin) and spectra analysis and backbone assignment were performed using CcpNmr v3 (Vranken et al. 2005; Stevens et al. 2011). Sidechain assignment was performed using NMRtist (https://nmrtist.org/) and manually validated using CcpNmr v3.

## 3. Results

### 3.1 Efficient NMR sample preparation using wheat-germ cell-free protein synthesis

Wheat-germ cell-free protein synthesis (CFPS) was used to prepare the NMR sample. This allowed rapid and efficient preparation of the protein domain, and provided sufficient amounts for NMR. Since it has been shown that supplementation with zinc ions significantly increases the solubility and stability of zinc-binding proteins (Matsuda et al., 2006; Jirasko et al., 2020), we investigated whether the addition of $ZnSO_4$ stabilizes the fold of the Gn zinc finger (ZF) domain by using circular

dichroism (CD) and the addition of EDTA to determine under which conditions the folded protein is obtained (Fig. S1). The CD spectra clearly show that the zinc-finger formation is EDTA-dependent and reversible, and that the fold appears more stable when $ZnSO_4$ is added. This shows that folded Gn$^{cyto}$ can be obtained in CFPS by simply adding $ZnSO_4$ to the reaction. Fig. 1 summarizes the sample preparation workflow, the pure protein preparation used to record the spectra, and the HSQC fingerprint spectrum obtained.



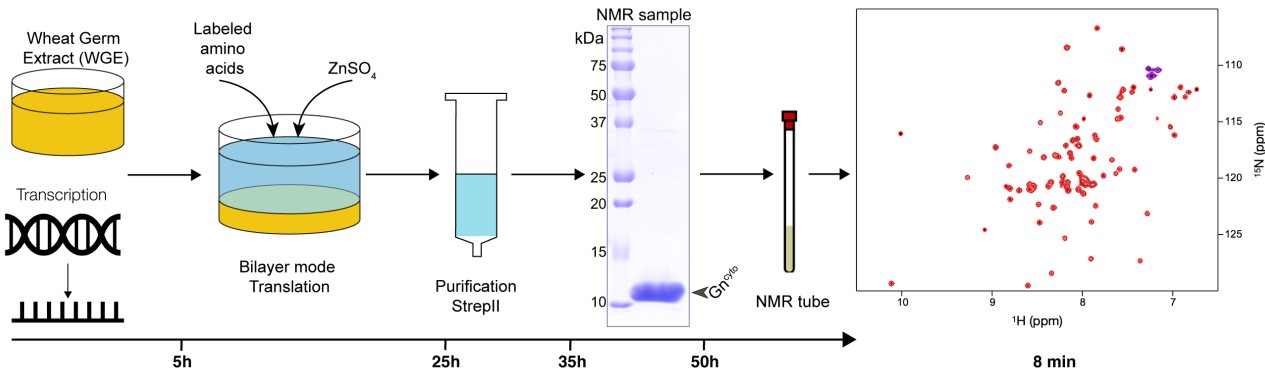


**Figure 1: Workflow for sample preparation.** Wheat germ extract and mRNA encoding the protein are used in a bilayer cell-free reaction. Addition of protonated or deuterated $^{13}$C/$^{15}$N labeled amino acids allowed synthesis of labeled protein. Zinc sulphate was added to the reaction to ensure correct zinc finger formation. After a single-step purification using the C-terminal Strep-II tag with concomitant buffer exchange, the protein showed high purity and was filled into the NMR tube. The entire sample preparation took approximately 2 days. The
SOFAST HN HSQC spectrum was acquired in 8 minutes.

## 3.2 NMR backbone assignments

The backbone assignments for the H$^N$, N$^H$, Cα, Cß, and Hα of Gn$^{cyto}$ have been reported in the literature (Estrada and De Guzman, 2011). In a first step, we analyzed the protein sample to confirm that the zinc finger fold was correctly obtained in the cell-free sample by comparing the NMR chemical shifts here obtained to previous data. To do this, we prepared a deuterated
sample, and recorded fingerprint and backbone correlation spectra, including $^1$H-$^{15}$N HSCQ, HNCA, HNCO, HNcoCA, and HNcoCACB. The HNCO was used to add the carbonyl assignments not previously determined. The HSQC spectra looked quite different at a first sight (Fig. S2), but this was mainly due to the use of different tags, and also slightly different strains. The remaining differences might be due to the small variation in pH (7 versus 6.45). The chemical shift differences (CSPs) are shown in Fig. S3. Large CSPs between previous assignments and our sample were observed for residues K732, T737, A746,
H752, K786 and I799. While the differences for the C-terminal I799 are likely due to the use of a different tag, the differences for the other residues remain unexplained. Most of the other CSPs remain below 0.2 ppm.



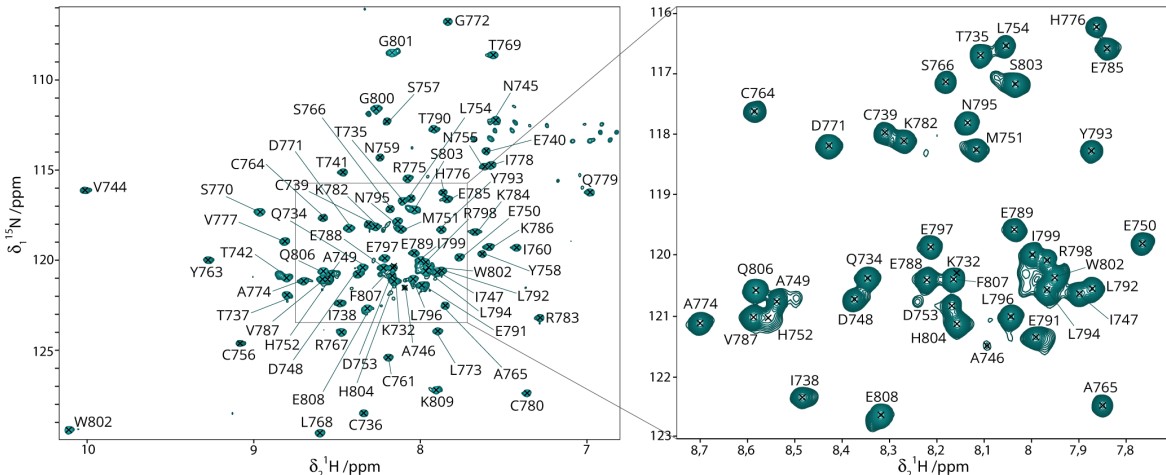

**Figure 2: HN backbone assignments.** HSQC spectrum with the assigned residues indicated. Assignments could mostly be transferred from the BMRB entry 17383 (Estrada and De Guzman, 2011). All assignments were verified using the NMR spectra allowing to identify sequential
backbone connections.

### 3.3 NMR side-chain assignments supported by NMRtist

Since we are ultimately interested in further analysis of the domain in its membrane-bound form using solid-state NMR, and in comparing the free and membrane-bound states of the domain, we set out to determine its chemical shifts on our construct, including the side-chain resonances missing in the previous work. To this end, we used NMRtist, an artificial
intelligence-based program that allows fully automated NMR spectrum assignment (and protein structure determination, (Klukowski et al., 2022)). NMRtist requires as input the sequence of the protein of interest, and peak lists of 2D and 3D assignment spectra. It is also possible to provide a partial assignment of the protein resonances. As input, we thus prepared peak lists of 2D hNH, and 3D HNCA, HNcoCA, CBCAcoNH, CBCAcoNH, HNcoCA, HCcH, and hCCH spectra.

Automated resonance assignments were tested with and without providing the software with protein backbone
assignments as input. Without providing any previous assignments, the software estimates the accuracy of chemical shift assignments at 81 %. This value increases to 89% when backbone chemical shifts are given as input. The assignment statistic of the latter NMRtist run is shown in Fig. 3.

**MAGNETIC**
**RESONANCE**
Open Access Discussions



**Figure 3: NMRtist output graph, F1 score and percentage of assigned expected peaks in the different spectra.** (A) Extent and reliability
of assignments obtained with the NMRtist automated resonance assignment algorithm using automatically picked peak lists. Each assignment
for an atom is represented by a blue rectangle. Light colors indicate an assignment classified as weak by the chemical shift consolidation.
(B) The box plots show the distribution of the F1 scores of the spectra, and the red dots represent the F1 scores calculated for the output of
the peak picking application call. Scores above the median (dotted line) indicate that the automated peak picking routine was able to process
a spectrum more accurately than 50% of the spectra in the benchmark. (C) Percentage of assigned peaks with respect to the expected number
of cross peaks in each spectrum, as given by the protein sequence.



Peak assignments were manually verified using the Peak Predict module of the CCPN v3 software, and comparing the predicted peaks with the actual spectra, particularly the hCH spectrum. Some residues required small manual shifting, or completion of missing resonances.



**Figure 4: ¹³C HSQC spectrum.** (A) Cα/Hα region with assignments shown on the spectrum. (B) Side-chain region, with crosses indicating assigned peaks. For assignments see BMRB submission.

We identified several causes that led to (minor) problems in NMRtist assignments. Firstly, the fact that 3D backbone and side-chain spectra were not recorded on the same sample led to some inaccuracies, as spectra on deuterated protein were



used as is. While the ¹⁵N- HSQC were still very similar and many assignments could be transferred, slight differences led to
increased errors. An additional error source was the relatively poor resolution of the HNcoCACB, which induced a greater
uncertainty in the Cα and Cβ chemical shifts in this spectrum. Most problems were confined to regions where peak overlap
occurred. Despite the few errors in the assignment, the use of NMRtist saved a lot of time, as all easy or isolated spin systems
were quickly assigned with high confidence. It also turned out that for the relatively small protein used and the high spectral
resolution, the backbone assignments did not help the program much, and the time needed for manual transfer of the
assignments thereof could have been saved.

In the end, most of the side-chain carbon and proton assignments could be achieved, as shown in the extracts of the ¹³C-HSQC
spectrum in Figure 4, where the assignments for the Hα/Cα are given, and all assigned side chains peaks are marked with a
cross. The data have been deposited at the BMRB under accession number 52372.

**4. Discussion and Conclusion**

We have shown here that the production and resonance assignment of small protein domains can be done in a highly efficient
manner by combining cell-free protein synthesis, NMR spectroscopy on remote high-field platforms, and automated resonance
assignment. The use of a high magnetic field not only provided the high resolution needed to assign side-chain protons, but
also the sensitivity needed to study the below 150 μM samples supplied by CFPS. While the sample preparation took about
two days, the acquisition of the side-chain assignment spectra took 2-3 weeks, and the analysis 1-2 weeks to verify the
automated assignments. The backbone assignments were not very time consuming since they had been previously established
(Estrada and De Guzman, 2011), but required verification, which took several days when done manually, but was not necessary
for automated assignments, which were similarly successful in the absence of this prior knowledge.

The protein considered here is rather small, with 69 amino acids plus a ten amino acid tag, and represents a subdomain of a
larger, membrane-bound protein. While solid-state NMR can address the membrane-bound forms, solution-state NMR spectra
of the isolated domains have the potential to accelerate a future analysis of the solid-state NMR 2D hCH and further 3D spectra,
such as hCCH. The knowledge gained here will also be used to analyze the differences between the isolated and membrane-
bound domain as highlighted by CSPs.

The number of situations where small domains can provide partial information on membrane-bound proteins is high in the
study of viral proteins, as many viral envelope proteins carry rather small ecto- or cytosolic domains that are easily accessible
to solution NMR. Together with AlphaFold2 predictions, sequential NMR resonance assignments can even obviate the need
for structure calculations if the models are of high quality, since the secondary chemical shifts from sequential assignments
provide immediate information on the secondary structure. The advantage of cell-free protein synthesis is that both membrane-
bound and isolated domains can be produced with similar efficiency using the same approach.



## Author Contributions

LB analyzed backbone spectra and prepared, together with ML, figures; ML, together with CM, manually verified, and finalized the side-chain assignments; LC and MLF prepared samples; AB wrote the manuscript, with input from all authors; LL recorded the NMR spectra and ran the NMRtist analysis; MC analyzed the spectra and supervised the project together with MLF.

## Acknowledgements

Financial support from the IR INFRANALYTICS FR2054 for conducting the research is gratefully acknowledged.

## Data availability

Chemical shifts are submitted to the BMRB with the accession number 52372.

NMR spectra have been deposited to https://doi.org/10.5281/zenodo.10938432.

## Competing Interest

AB is Associate Editor with MR.

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
