# Peer review of "NMR side-chain assignments of the Crimean-Congo haemorrhagic fever virus glycoprotein n cytosolic domain"

_Magnetic Resonance, 2024_

## Author Comment (AC1)

Dear anonymous referee,

we would like to thank you very much for your very careful reading of the manuscript and your interesting remarks. Please find below our replies.

A. Böckmann for all authors

The use of NMRtist in this manuscript adds very little scientifically as it is consistent with the reported and expected performance of the software and while it is noteworthy, its significance appears overstated.

*-> We have no problem to tune down the significance statement*

Figure 3 in particular appears to be some internal reporting from the software rather than novel analysis of the performance of that software (if it is a novel analysis it should be described in more detail). This figure may be suitable to deposit as supplementary data but does not add to the presented work in a meaningful way.

*-> This is indeed not novel, but is the standard output from the software reporting on the assignments done by the program. It shows how the program has used the spectra and the result of the run. We had reported this type of output in a previous paper (Schmidt, Elena, Julia Gath, Birgit Habenstein, Francesco Ravotti, Kathrin Szekely, Matthias Huber, Lena Buchner, Anja Böckmann, Beat H Meier, et Peter Güntert. « Automated Solid-State NMR Resonance Assignment of Protein Microcrystals and Amyloids ». Journal of Biomolecular NMR 56, no 3 (mai 2013): 243-54. https://doi.org/10.1007/s10858-013-9742-x.), and we would have thought that this is good practice. If the referee has a strong opinion about this, we can certainly move it to the SI.*

Similarly the abstract speak of remote access, but the manuscript does not detail how this is done or what is novel about the remote access that makes this of note in this publication (beyond perhaps a reference to NMRlib).

*-> We can remove "remote" and insert NMRlib (the reference is actually present but was unfortunately not formatted in the reference list).*

Major revision required:

1. The differences in chemical shift around residue 752 is a little concerning as this coordinates a Zn ion. I'm concerned that the Zn is not saturated. It is noted that a concentration of 0.1 mM of zinc sulfate is used, as was present in the sample used in the previously published work, but it is unclear if this is sufficient to saturate the Zn binding. The published work reported that a 1 mM of zincsulfate concentration was used during purification when the sample was dilute and the sample subsequently concentrated, with the final buffer containing zinc sulfate at a concentration of 0.1 mM. Thus, in the reported scenario the protein has been exposed to saturating

concentrations of Zn and likely maintains this throughout. But in this work it is unclear if the addition of 0.1 mM ZnSO4 during the processing is sufficient given that protein concentrations are much higher during CFPS. It would be important to perform a titration of ZnSO4 to the protein (monitored by 15N HSQC) to determine if the chemical shift difference remain when the sample is in the presence of saturating Zn concentrations.

*-> In the literature, cell-free protein synthesis in dialysis mode was done in presence of 100 uM (Matsuda et al., 2006). We ourselves have worked before with a Zinc-binding protein produced in a very similar cell-free set-up, which yielded excellent solid-state NMR spectra of a well-folded protein (Jirasko et al., Angew. Chem. 2020). In the present report, we have worked throughout with buffers containing 100 uM Zn ions throughout all steps, and in absence of EDTA. The concentration of the protein in the cell-free reaction (here done in bilayer mode) was 46 uM, and thus not higher than during purification or in the final NMR sample.*

*-> Furthermore, the 15N HSQC spectrum shown in Estrada et al. 2011 does not actually show a peak at the chemical shifts given for His752 (coordinating ZF1). We therefore believe that this is rather an unfortunate misclassification. Indeed, the cysteine residues of ZF1 clearly show the chemical shifts described by Estrada et al. 2011, confirming the proper formation of the zinc finger.*

2. It is not entirely clear why the deuterated sample was produced and when it was used. It is noted that the protonated sample was used to transfer assignments, but the sidechain assignments include proton detected hCCH experiments which suggests that the deuterated sample was not used for this. It would be helpful if the results section included a listing of which sample was used for which experiment. Ideally included a clear rationale why deuteration is required at all.

*-> As is often the case, the reason is historical. Our first goal in this project was to confirm the assignments of Estrada et al. 2011, since we observed discrepancies with our solid-state NMR spectra. So we used a deuterated sample. We then realised that it would be interesting to also obtain side-chain assignments including protons, as we wanted to understand the proton line widths observed in the solid-state NMR spectra. So we made a protonated sample to do this. As we use spectrometer time on the National Access Programme, we found it exaggerated to ask to record all the sequential spectra on this sample again.*

Minor:

It is noted that Europe is a non-endemic region for this virus but the endemic regions have not been defined. I appreciate this may be apparent from the name, but names of viruses are not always very descriptive of their origin or endemic regions.

*-> CCHFV is endemic in most parts of Africa, in the Balkans, in the Middle East and in Asia (Shahhosseini et al., 2021)*

Latin names such as Hyalomma should be italicised.

*-> We can addressed this.*

The note of an outbreak in France needs a reference or should be removed.

*-> We did not refer to an outbreak, but to the first detection of the virus in France. We will insert the reference (Bernard, C., Joly Kukla, C., Rakotoarivony, I., Duhayon, M., Stachurski, F., Huber, K., Giupponi, C., Zortman, I., Holzmuller, P., Pollet, T., Jeanneau, M., Mercey, A., Vachiery, N., Lefrançois, T., Garros, C., Michaud, V., Comtet, L., Despois, L., Pourquier, P., Picard, C., Journeaux, A., Thomas, D., Godard, S., Moissonnier, E., Mely, S., Sega, M., Pannetier, D., Baize, S., and Vial, L.: Detection of Crimean–Congo haemorrhagic fever virus in Hyalomma marginatum ticks, southern France, May 2022 and April 2023, Eurosurveillance, 29, https://doi.org/10.2807/1560-7917.ES.2024.29.6.2400023, 2024).*

*.*

In the discussion, the statement that the high field provided the needed sensitivity and resolution is not supported by the data. Clearly it would help, but it is not shown that it is needed, and is unlikely to be the case for such a small protein. Perhaps replace needed with facilitated or similar.

*-> We can do this.*

---

## Author Response (AR1)

*Dear referees,*

*we would like to thank you very much for your very careful reading of the manuscript and your interesting remarks. Please find below our replies.*

*Morgane Callon for all authors*

Referee 1, first round:

The use of NMRtist in this manuscript adds very little scientifically as it is consistent with the reported and expected performance of the software and while it is noteworthy, its significance appears overstated.

We have tuned down the significance statement

Figure 3 in particular appears to be some internal reporting from the software rather than novel analysis of the performance of that software (if it is a novel analysis it should be described in more detail). This figure may be suitable to deposit as supplementary data but does not add to the presented work in a meaningful way.

We have moved to the SI.

Similarly the abstract speak of remote access, but the manuscript does not detail how this is done or what is novel about the remote access that makes this of note in this publication (beyond perhaps a reference to NMRlib).

We removed "remote" and inserted the NMRlib reference (the reference was actually present but was unfortunately not formatted in the reference list).

Major revision required:

1. The differences in chemical shift around residue 752 is a little concerning as this coordinates a Zn ion. I'm concerned that the Zn is not saturated. It is noted that a concentration of 0.1 mM of zinc sulfate is used, as was present in the sample used in the previously published work, but it is unclear if this is sufficient to saturate the Zn binding. The published work reported that a 1 mM of zincsulfate concentration was used during purification when the sample was dilute and the sample subsequently concentrated, with the final buffer containing zinc sulfate at a concentration of 0.1 mM. Thus, in the reported scenario the protein has been exposed to saturating concentrations of Zn and likely maintains this throughout. But in this work it is unclear if the addition of 0.1 mM $ZnSO_4$ during the processing is sufficient given that protein concentrations are much higher during CFPS. It would be important to perform a titration of $ZnSO_4$ to the protein (monitored by 15N HSQC) to

determine if the chemical shift difference remain when the sample is in the presence of saturating Zn concentrations.

In the literature, cell-free protein synthesis in dialysis mode was done in presence of 100 uM (Matsuda et al., 2006). We ourselves have worked before with a Zinc-binding protein produced in a very similar cell-free set-up, which yielded excellent solid-state NMR spectra of a well-folded protein (Jirasko et al., Angew. Chem. 2020). In the present report, we have worked throughout with buffers containing 100 uM Zn ions throughout all steps, and in absence of EDTA. The concentration of the protein in the cell-free reaction (here done in bilayer mode) was 46 uM, and thus not higher than during purification or in the final NMR sample.

Furthermore, the 15N HSQC spectrum shown in Estrada et al. 2011 does not actually show a peak at the chemical shifts given for His752 (coordinating ZF1). We therefore believe that this is rather an unfortunate misclassification. Indeed, the cysteine residues of ZF1 clearly show the chemical shifts described by Estrada et al. 2011, confirming the proper formation of the zinc finger.

1.  It is not entirely clear why the deuterated sample was produced and when it was used. It is noted that the protonated sample was used to transfer assignments, but the sidechain assignments include proton detected hCCH experiments which suggests that the deuterated sample was not used for this. It would be helpful if the results section included a listing of which sample was used for which experiment. Ideally included a clear rationale why deuteration is required at all.

As is often the case, the reason is historical. Our first goal in this project was to confirm the assignments of Estrada et al. 2011, since we observed discrepancies with our solid-state NMR spectra. So we used a deuterated sample. We then realised that it would be interesting to also obtain side-chain assignments including protons, as we wanted to understand the proton line widths observed in the solid-state NMR spectra. So we made a protonated sample to do this. As we use spectrometer time on the National Access Programme, we found it exaggerated to ask to record all the sequential spectra on this sample again.

Minor:

It is noted that Europe is a non-endemic region for this virus but the endemic regions have not been defined. I appreciate this may be apparent from the name, but names of viruses are not always very descriptive of their origin or endemic regions.

CCHFV is endemic in most parts of Africa, in the Balkans, in the Middle East and in Asia (Shahhosseini et al., 2021). We added.

Latin names such as Hyalomma should be italicised.

We addressed this.

The note of an outbreak in France needs a reference or should be removed.

We did not refer to an outbreak, but to the first detection of the virus in France. We will insert the reference (Bernard, C., Joly Kukla, C., Rakotoarivony, I., Duhayon, M., Stachurski, F., Huber, K., Giupponi, C., Zortman, I., Holzmuller, P., Pollet, T., Jeanneau, M., Mercey, A., Vachiery, N., Lefrançois, T., Garros, C., Michaud, V., Comtet, L., Despois, L., Pourquier, P., Picard, C., Journeaux, A., Thomas, D., Godard, S., Moissonnier, E., Mely, S., Sega, M., Pannetier, D., Baize, S., and Vial, L.: Detection of Crimean–Congo haemorrhagic fever virus in Hyalomma marginatum ticks, southern France, May 2022 and April 2023, Eurosurveillance, 29, https://doi.org/10.2807/1560-7917.ES.2024.29.6.2400023, 2024).

In the discussion, the statement that the high field provided the needed sensitivity and resolution is not supported by the data. Clearly it would help, but it is not shown that it is needed, and is unlikely to be the case for such a small protein. Perhaps replace needed with facilitated or similar.

Done.

**Referee 1, second round:**

Thank you for your response. In relation to Figure 3. As I noted in my review, it is unclear how the interpretation of this will add value for the reader. Unless the values and the graphs provide information that is important for the interpretation of the results or to support a claim of the manuscript it is an important technical detail but not a noteworthy result in itself to be presented as part of the main manuscript. The general discussion of the discrepancies found is useful and of note for those wishing to understand the performance of NMRtist, but since all assignments were validated manually anyway, the uncertainties in the initial assignments bear little impact on the results presented. I would see this as a report of the tools used and not a result of this investigation and would suggest placing this in the supplementary materials.

We moved it to the supplementary materials

Thank you for the clarification of the assignment of His752. It is important to note this in the manuscript (a good reminder of the importance of inclusion of primary data in an accessible database as the editor requested). My concern however is that even if the protein is at 46 micromolar and the zinc concentration is at a 100 micromolar, since the protein has 2 zinc binding sites it is barely saturated. It is likely that even if it is not saturated during expression that it may be exposed to sufficient amounts of zinc to become saturated during the strep purification, but since the details of this purification are not provided (wash buffer volume, elution buffer volume) it is hard to be certain. A simple experiment would be to add 1 mM zinc

sulfate to one of the samples (or a fresh 15N labelled sample) and to demonstrate that the zinc binding is saturated. As you have noted there is a surprising difference between the published HSQC and the one you are observing, and it would be prudent to rule out the most obvious cause of this which would be zinc binding, given the importance of this for the folding of the protein. Similarly, as you note this may be due to the difference in pH which is unfortunately very close to the pKa of histidine and may have caused a change in the affinity of the protein for zinc at one of the metal binding sites. The source of the difference is otherwise not resolved.

We now added 1 mM Zinc sulfate as a final concentration to the protonated Gn$^{cyto}$ sample and recorded a 15N-HSQC spectrum. No differences can be observed between the spectra taken before and after the addition of Zinc sulfate in large excess, showing clearly that the Zn binding sites are saturated. We added this spectrum to the SI.

 I note also that there are number of unassigned peaks in the new spectrum, are these from impurities or minor/unfolded states (in Figure S2 there is a prominent peak between 759 and 790 which is much less intense in Figure 2 - are these from different samples)?

The two experiments are different and recorded on different samples: the 15N-HSQC in Fig 2 is a BEST-TROSY experiment recorded on the protonated sample and the 15N-HSQC in Fig S3 (previously S2) is a SOFAST experiment recorded on the deuterated samples. We added the information in the manuscript.

Several of the unassigned peaks belong to Desthiobiotin used during the protein purification and residual glycerol (from centrifugal concentrator). We annotated them in the $^{13}$C-HSQC spectrum. Some of the other unassigned peaks belonged to the N-terminal Met or the tag, and we added them.

Finally, given the excellent completeness of the assignment it should be trivial to predict the secondary structure of the protein using a tool like TALOS. I would suggest you include this and compare the secondary structure elements to that of the published (or AlphaFold predicted) structure. This would add further confidence that the folding is maintained.

We compared the secondary structure predicted by TALOS-N with that of the published NMR structure (PDB 17383). The secondary structure elements match, confirming that the folding of Gn is maintained. We added the Figure to the SI.

Finally, I appreciate the historical reasons, but a listing of which sample is used for which experiments should be clearly presented in the experimental or results section.

We added a Table in SI showing which experiment was recorded on which sample.

**Referee 2:**

The manuscript by Brigandat et al. presents NMR resonance assignments (including side chain $^1H$ and $^{13}C$) of a short (69 residue long) soluble domain from a viral envelop protein using standard 3D solution NMR experiments. The authors also put forward that the results have been obtained by combining cell-free protein synthesis, standardized NMR pulse sequences, and automatic (but manually verified and completed) data analysis.

I completely agree with all comments and concerns made by anonymous Reviewer 1, and I will thus not repeat these points in my review.  My overall impression is that the original results presented in this short communication concern essentially the NMR resonance assignments, and that the various experimental approaches used to obtain them can be considered as quite standard these days. It is stated in the journal's publication guidelines: "Routine applications of established techniques and minor technical advances are considered to be outside its scope". Therefore, I am not convinced that the journal "Magnetic Resonance" is the appropriate place for publishing this work, but rather feel that this manuscript fits perfectly to "Biomolecular NMR Assignments", a journal that is entirely devoted to publishing and disseminating new NMR assignments of proteins (and nucleic acids).

In the last five years, Anja Böckmann's group has published very few non-open access (OA) articles, recognizing the importance of open science. A notable exception is an assignment note in J BioNMR Assign. With the launch of MR, the NMR community has finally introduced an open access journal. It would be a pity, in our view, if the assignment notes, which we believe are an important part of the recognition that students and post-docs receive for their work in biomolecular NMR, were to be removed from this endeavor and their contribution no longer recognized.

Minor point:

- Page 4 (figure caption 1): a SOFAST HN HSQC experiment does not exist !

Thanks, we corrected this

**Referee 3:**

The work by Brigandat et al. reports on the sidechain resonance assignment of a small protein, completing the already published backbone assignment. The methods employed by the authors are well established. Therefore, the novelty lies entirely in the assignment, and the "significant innovation" component is questionable. Nonetheless, the work provides valuable information. Publishing the contribution in 'Biomolecular NMR Assignments' would be more appropriate.

See reply to comment to anonymous referee 2. J Biomol NMR assignments has served the community for a long time, and the group of A. Böckmann has published many contributions to this journal, which was created to provide a space for this important, but not always very innovative and spectacular results. A problem for the journal has always been that access to it was limited to a few institutions, and that pay-OA was terribly expensive. In the era of open science, and with a new OA player which finally caters the MR community in all MR aspects, we prefer to switch today to MR in order to comply with the OA rules of many granting agencies, but also to make our work accessible to a wider community.

Comments:

Abstract and Introduction:

- Please clearly state the number of protein amino acid residues (69 aa) in the abstract and introduction.

We included this.

- Line 13: Clarify the meaning of "remote access."

 It means that the spectrometer was operated from Lyon, whereas it is located in Grenoble. We removed this statement as it seems to be a fairly common possibility today.

Protein Expression and Purification:

 - Explain why the streptavidin tag was not cleaved after purification.

The cleavage of the tag would have required an additional purification step that would have reduced the protein yield. As the peaks belonging to the tag can be assigned in the hNH spectra, we did not cleave the tag. Furthermore, the chemical shifts do not seem to be much affected except for the residue just before the tag (I799).

- Clarify the rational for using protease inhibitors late in the purification process instead of in the early steps.

There is no significant protease activity in the WG extract (see Fogeron et al. Front. Mol. Biosci. 2021). We added protease inhibitor afterwards to avoid degradation of the protein during NMR measurements.

 - I recommend performing size exclusion chromatography as a final purification step to remove potential protein aggregates or nucleic acids.

Thank you for the suggestion. Since we did not observe aggregates or nucleic acids in the spectra, we did not implement such a step.

 - Consider using a buffer other than phosphate, as phosphate can precipitate in the presence of Zn ions.

Thank you for the interesting remark. We did not observe any precipitation here, but we will consider this if needed.

 - Line 97: Specify whether "strain" refers to viral or bacterial expression strain. Clearly outline the differences between the two protein constructs mentioned.

We added the protein sequences with the differences highlighted in the SI, Fig. S3. We have made this clearer in the main text.

 - Line 98: If completing an already published assignment, explain why experimental conditions from the previous work were not used.

Our aim was not to complete a previously published assignment, but to obtain assignments that would facilitate the study of the membrane-bound form. We therefore used the experimental conditions that are closest to those we use when studying membrane proteins inserted into lipids.

- Line 139 "help the program much": Provide quantitative details.

We quantified it in the manuscript.

- Figure 3: caption "indicate that the automated peak picking routine was able to process a spectrum more accurately than 50% of the spectra in the benchmark": Please clarify "process" and "benchmark", or provide the reference from where it was taken from (https://nmrtist.org/static/public/examples/ARTINA/ARTINA_dataset3.html).

We added the reference.

---

## Author Response (AR2)

Dear Editor, dear Prof Otting,

we would like to thank you for the careful reading of the manuscript and the comments. Please find below our answers.

Morgane Callon on behalf of all authors

When reading through the revised manuscript to compare with the referees' comments, I came across a few points that deserve addressing.

Figure 3: A and B labels missing in figure.
Thank you, we added them.

Figure S2: please report the concentration of the protein.
The protein concentration in Figure S2 was 60 μM. We added this information to the manuscript.

Line 57: please report the volume of the cell-free reaction.
The volume of the cell-free reaction is in total 36 mL (6 x 6 mL wells). We added this information to the manuscript.

Line 60: please report the (presumably commercial) source of amino acids.
The amino-acids are purchased from Cambridge Isotope Laboratories. We added this information to the manuscript.

Line 64: please specify the protease inhibitor used.
We used the cOmplete, EDTA-free protease inhibitor Cocktail (Roche). We added this information to the manuscript.

Line 68: please clarify, whether both spectrometers were equipped with a cryoprobe.
The two spectrometers are equipped with cryoprobe. We added this information to the manuscript.

Line 79: please clarify whether CFPS was used for one or two NMR samples.
All samples were produced using CFPS. We clarified this in the manuscript.

Line 84: Fig. S1 does not prove that zinc-finger formation is reversible (which would require the recovery of the CD spectrum following the addition of zinc after having extracted zinc with EDTA).
We now provide CD data showing that the zinc-finger formation is reversible, as already shown by Estrada et al. We could recover the CD spectrum after addition of zinc after its removal with EDTA. We added this Figure as Panel B to Figure S1.

Line 86: please spell atom names with small characters.
Thank you, we modified it in the manuscripts

Line 101: the spectral changes relative to the originally published ones are unexpectedly large. The explanation given that this is due to the use of different tags is not convincing. The structure 2L7X indicates that the C-terminal peptide is disordered and far from K732, T737 and A746, which show large changes in chemical shifts compared to the previous assignments (Fig. S4). The C-terminal peptide may well be very flexible (Fig. 2 indicates chemical shifts characteristic of random coil conformations and Fig. S6 shows CSPs characteristic of flexible C-termini). The C-terminal tag is unlikely to change chemical shifts on the opposite face of the folded protein domain. Arguably, the difference between different viral strains does not offer a plausible explanation either, as I779 is reported to be the only residue in the structured part of the protein core different between this construct and the one of the previous study. There is no evidence that this mutation changes the chemical shifts of the neighbouring residues.

One key factor in our decision to conduct a comprehensive assignment in-house were the discrepancies observed in the chemical shifts previously published by Estrada et al. Below we show some discrepancies between Figure 3A in their paper and their deposited BMRB files. Besides T737 chemical shift errors, neither A746 nor H752 are annotated anywhere in the 2D spectrum. Further annotation errors are shown in the last panel, and other issues were observed regarding carbon chemical shifts. We believe this data speaks for itself. Still, it is out of the scope of our manuscript to provide proof with respect to previous erroneous findings.

[Figure]

[Figure]

T737:

| 39 | . 1 1 | 9 | 9 | THR | H  | H  | 1  | 9.553   | . . 1 . . . . | 37 | THR | HN | . 17383 1 |
| 40 | . 1 1 | 9 | 9 | THR | HB | H  | 1  | 4.064   | . . 1 . . . . | 37 | THR | HB | . 17383 1 |
| 41 | . 1 1 | 9 | 9 | THR | CA | C  | 13 | 63.839  | . . 1 . . . . | 37 | THR | CA | . 17383 1 |
| 42 | . 1 1 | 9 | 9 | THR | CB | C  | 13 | 69.111  | . . 1 . . . . | 37 | THR | CB | . 17383 1 |
| 43 | . 1 1 | 9 | 9 | THR | N  | N  | 15 | 121.780 | . . 1 . . . . | 37 | THR | N  | . 17383 1 |

A746:

```
83 . 1 1 18 18 ALA H   H 1    8.841 . . 1 . . . .   46 ALA HN  . 17383 1
84 . 1 1 18 18 ALA HA  H 1    4.176 . . 1 . . . .   46 ALA HA  . 17383 1
85 . 1 1 18 18 ALA CA  C 13  55.854 . . 1 . . . .   46 ALA CA  . 17383 1
86 . 1 1 18 18 ALA CB  C 13  18.389 . . 1 . . . .   46 ALA CB  . 17383 1
87 . 1 1 18 18 ALA N   N 15 121.144 . . 1 . . . .   46 ALA N   . 17383 1
```

**H752:**

[Figure]

[Figure]

[Figure]

```
112 . 1 1 24 24 HIS H   H  1    8.068 . . 1 . . . .  52 HIS HN  . 17383 1
113 . 1 1 24 24 HIS HA  H  1    4.739 . . 1 . . . .  52 HIS HA  . 17383 1
114 . 1 1 24 24 HIS CA  C 13   60.871 . . 1 . . . .  52 HIS CA  . 17383 1
115 . 1 1 24 24 HIS CB  C 13   29.995 . . 1 . . . .  52 HIS CB  . 17383 1
116 . 1 1 24 24 HIS N   N 15  120.999 . . 1 . . . .  52 HIS N   . 17383 1
```

While these chemical shift differences are not the key message of this manuscript, the explanation for the chemical shift differences between old and new constructs remains unsatisfactory.
We hope that the above analysis provides a sufficient explanation.
Were the previously published assignments established using samples prepared without the addition of zinc during purification whereas the present assignment was of protein with zinc supplied during sample preparation? If so, the CD spectra of Fig. S1 seem to suggest that this could have a substantial impact on the structure.
In our experiments, the zinc is added during sample preparation, as also in the previously published assignment. The new CD curve showing that the zinc-finger formation is reversible, this would anyway not have an impact on the structure of the protein.

Please clarify whether the spectra of Fig. S2 were recorded in the presence of 50 mM phosphate – as referee 3 points out, zinc phosphate is insoluble (in fact, as Wikipedia tells, it is used as dental cement). The authors should either provide more convincing data (NMR spectra with different amounts of zinc recorded in the absence of phosphate) …
The previously published spectrum in Figure 2S (Estrada et al., black spectrum) was recorded in 10 mM phosphate buffer while our spectrum (red spectrum) was

recorded with 50 mM phosphate buffer. However, in our CD experiments performed with 50 mM phosphate, we do not observe any precipitation.

… or give an acceptable explanation. An experimental resolution of the issue would significantly raise the significance of this manuscript.

Considering our above annotations of the published data, this manuscript could only become more relevant if we could show that the 3D structure described in the literature does not comply with the erroneous chemical shifts. As Figure S6 shows, the secondary structure elements are not much impacted by the differences. A quick check on NOEs used shows that mainly intra-residue, and i,i+-1,2 were used for 737 and 752. A comprehensive and solid verification of previous data is outside the scope of our manuscript.

Line 109: A more helpful sentence may read "All resonance assignments of backbone amides were verified by 3D NMR spectra recorded of the 2H-13C-15N labeled sample."
Thank you for the correction. We changed it in the manuscript.

Line 116: clearer wording would be 'amino acid sequence'.
Thank you, we corrected it in the manuscript.

Line 117: "assignment spectra" is unfortunate lab jargon.
We changed it.

Line 125: "completion of missing resonances" – please spell out more accurately.
We modified the sentence.

Line 133: "as is" appears superfluous.
We removed the end of the sentence.

Line 134: "15N-HSQC spectra". "increased errors": increased relative to what?
We are here referring to errors due to the different labeling (protonated vs deuterated) of the two samples, as deuteration leads to shifts of the CS. We clarified this in the manuscript.

Line 140: please report the correct DPB ID. Is it 2L7X?
Thanks - yes, it is 2L7X. We corrected in the manuscript.